# Well-Being of High-Level Managers during the Pandemic: The Role of Fear of Negative Appearance, Anxiety, and Eating Behaviors

**DOI:** 10.3390/ijerph20010637

**Published:** 2022-12-30

**Authors:** Anna Hryniewicz, Dominika Wilczyńska, Daniel Krokosz, Konrad Hryniewicz, Mariusz Lipowski

**Affiliations:** 1Faculty of Physical Education, Gdańsk University of Physical Education and Sport, 80-336 Gdansk, Poland; 2Department of Marketing and Quantitative Methods, Faculty of Management and Quality Science, Gdynia Maritime University, 81-225 Gdynia, Poland

**Keywords:** well-being, managers, eating attitudes, body image, COVID-19

## Abstract

(1) Background: The COVID-19 pandemic has caused unprecedented changes in the contemporary world, significantly affecting the work of companies, especially management staff. This study investigated whether fear about one’s health (caused by the pandemic, disordered eating attitudes, or concerns about one’s body image) has a negative relationship with the well-being of managers. (2) Methods: *N =* 354 managers (222 women, 126 men, and 6 people with no gender identity) participated in the study. The following psychometric instruments were used: the psychological well-being scale, the coronavirus anxiety scale, the fear of negative appearance evaluation scale, and the eating attitude test-26. Results: the fear of negative appearance influenced the well-being of the studied managers. However, this relation was mediated by dieting as well as bulimia and food preoccupation. (4) Conclusions: the well-being level depended on the managers’ positive body images, but only when mediated by healthy dieting and eating attitudes. While the well-being level of managers was high, it is worth further exploring how they can flourish and develop in life and work, which can also transfer to the quality of life of their co-workers and companies. However, the subject of the well-being of managers warrants more research; for example, by considering different moderators, such as job experience, gender, and age. Moreover, experimental studies examining the effectiveness of different interventions for the physical and mental health of managers could be worth investigating.

## 1. Introduction

At the beginning of 2020, the COVID-19 pandemic caused significant changes in the functioning of entire societies. The hitherto unknown virus (associated with acute pneumonia) constituted a serious threat, especially for people suffering from chronic diseases. The novelty of the situation was supplemented with an inconsistent information policy on the part of the media and national governments, which further complicated an already difficult situation. When the WHO [1] announced that the coronavirus constituted a global pandemic in March 2020, many governments announced lockdowns, limiting the reasons for which people could leave their homes and reducing traveling to work to a minimum. People in contact with infected persons had to quarantine to limit the spread of the virus.

During the pandemic, workplaces had to make significant changes in a very short timeframe—many companies switched to remote working, using online communication platforms, email, and intranet [2]. Many employees, especially in the catering and hotel industries [3], were forced to leave their jobs or stop working, and, where possible, employees performed their tasks from home [4]. According to Gartner [5], in half of the world’s companies, about 80% of employees during the pandemic worked from home. This situation was a source of new difficulties and challenges for managers, potentially affecting their well-being. Introducing changes in work organizations and supervising employees working from home were undoubtedly factors that, in addition to traditional duties, increased the workloads of managers. [6,7,8,9]. Lower levels of well-being among managers, even before the pandemic, were usually associated with increased levels of stress due to the duties and requirements associated with their work [10,11,12]. Concern for one’s health (and that of one’s relatives) often contributed to a significant increase in anxiety and a deterioration in mental health [13]. Although the media has mainly paid attention to physical COVID-19-related health concerns, many studies have shown deterioration in mental health, particularly depression and anxiety disorders [14,15,16]. It should also be emphasized that managers can often serve as role models for employees, especially regarding healthy lifestyles [17]. Understanding the relationship between healthy eating and well-being in managers can contribute to a better understanding of the determinants of a healthy lifestyle in employees. People who experienced healthy lifestyles, took care of their diets, and were physically active before the pandemic, may have had difficulties continuing these practices during the COVID-19 pandemic, which resulted in the recent worldwide increase in body weight. This phenomenon is a new pandemic, termed covibesity [18]. Given the significant impact of social media on the perceptions of physical attractiveness and one’s own body, the discomfort associated with covibesity could have had a substantial impact on people’s well-being during the pandemic. Research shows that due to spending substantial amounts of time on social media platforms, many people have been observing their images for longer amounts of time than prior to the advent of social media [19]. Research shows that one-third of people who concentrate on their images for long periods experience decreased satisfaction with their appearances [20].

Furthermore, difficulties participating in daily activities and negative perceptions of one’s own appearance may have caused significant changes in diets, which may be associated with eating disorders [21]. As Scharmer et al. [22] pointed out, people with a high intolerance to uncertainty were particularly at risk of developing eating disorders during the pandemic.

Managers are a professional group that may have been particularly vulnerable to the consequences of the COVID-19 pandemic. Significant sources of anxiety and stress for this group included feelings surrounding health threats, stress about work, and the need to reorganize their workplaces for subordinates Numerous studies [23,24] indicate a significant relationship between anxiety and mental health consequences, especially in the forms of eating disorders and negative attitudes about one’s appearance. The aim of the present study was, therefore, to review how anxiety surrounding the COVID-19 pandemic was associated with eating disorders and self-image assessment, and how these factors were associated with the sense of well-being of managers. These factors, in addition to their significant impact on well-being, are directly related to healthy lifestyles.

Given that the pandemic was unprecedented and that previous scientific research did not produce coherent theories on how high-level workers could respond to this situation, this study is exploratory. Therefore, the following research questions were posed: (1) How did the fear of COVID-19 and the fear of negative image predict the life satisfaction feelings in managers? (2) Are eating behaviors mediators in this relationship?

The next section presents the methods and describes the group of participants. The third section presents the results, followed by a discussion, study limitations, and future directions.

## 2. Materials and Methods

### 2.1. Sample

Managers from Polish companies (mainly IT and insurance companies) were invited to participate in the study using the snowball sampling procedure. Data were collected in 2021 and 2022 during the 4th and 5th waves of the pandemic. The study was conducted in the form of an online questionnaire. Prior to the start of the study, the participants gave written consent to participate.

The study involved *N* = 354 managers (*n* = 222 women; *n* = 126 men, and *n* = 6 people who did not answer the question about gender). The respondents held managerial positions for *M* = 12.33 years (*SD* = 8.65). The majority of respondents (58%) were top-level managers, 25% were mid-level, and 17% were line managers. Each participant had to manage at least 10 employees to be included in the study. The mean age of the participants was *M* = 48.14 (*SD* = 9.69).

### 2.2. Instruments

The following research instruments were used in this study: the psychological well-being scale was used to determine the outcome variable; the coronavirus anxiety scale and the fear of negative appearance evaluation scale were used to determine the predictors; and the eating attitude test was used to determine the mediator. Below is a detailed description of the instruments.

The psychological well-being scale (PWBS) [25] was used (i.e., its Polish version by Karaś and Cieciuch) [26]. Participants responded via a six-point Likert scale (1—*strongly disagree*; 6—*strongly agree*) and subscales pertaining to self-acceptance, environmental mastery, positive relations, purpose in life, personal growth, and autonomy (e.g., “*I’m pretty good at dealing with the responsibilities of everyday life*”; “*I like most of my character traits*”). Each scale consists of three test items. The higher the score, the higher the rated well-being level. Cronbach’s alpha (the Polish version of the scale) was moderate (0.77).

In order to measure levels of anxiety during the pandemic, the coronavirus anxiety scale (CAS) [27] was used, i.e., the Polish language version developed by Skalski et al. [28]. The questionnaire consists of five items (e.g., “I felt paralyzed or frozen when I thought about or was exposed to information about the coronavirus”), to which the participants responded on a five-point Likert scale (1—*not at all*; 5—*nearly every day for the past two months*). Higher CAS scores indicate dysfunctional anxiety associated with COVID-19. Cronbach’s alpha (the Polish version of this scale) was 0.80.

The fear of negative appearance evaluation scale (FNAES) [29] was used to determine levels of concerns related to body image (e.g., ”I am concerned about what other people think of my appearance”; “I am afraid other people will notice my physical flaws”). The scale has been translated into Polish [23]. The participants responded to six items regarding negative appearance problems on a five-point Likert scale (1—*not at all*; 5—extremely). The psychometric properties of the scale were very good, with Cronbach’s alpha (the Polish version) reaching 0.95.

To assess the eating behaviors of participants, we used the eating attitude test (EAT-26) [30], i.e., its Polish version by Rogoza, Brytek-Matera, and Garner [31]. The test consists of three subscales concerning: (1) dieting, (2) bulimia and food preoccupation, and (3) oral and control behavior (e.g., “I have the impulse to vomit after meals”; “I am terrified about being overweight”). The participants responded to 26 items on a 5-point Likert scale (1—*always*; 6—*never*). The psychometric properties of the Polish version of the EAT-26 were relatively good (Cronbach’s alpha = 0.85).

## 3. Results

In order to verify the estimates of the theoretical model, structural equation modeling (SEM) was used. To calculate the SEM models, we worked in the R program and used the “lavaan” package [32]. The model’s calculations were based on the MLR algorithm (maximum likelihood estimation with robust Huber–White standard errors). This method allows the computation of robust estimates and standard errors [33]. Table 1 presents basic statistics related to the variables examined. These results show that the measured variables were reliable α > 0.70 (the exception was the oral control measurement: α = 0.59), variable, and significantly correlated *p* < 0.05.

### 3.1. Model Fit and Model Comparisons

Table 2 shows a comparison of the three models. The first comparison was between a saturated model and a model constrained to zero covariances between independent variables, the second one was between a saturated model and one constrained to zero covariances between mediator variables, and the third between one constrained to zero covariances between independent variables and another constrained to zero covariances between mediators. These comparisons suggest that the constrained models had a worse fit than the saturated one and that the constrained mediators model had a poorer fit than the constrained independent variables model. In the text, we reported the saturated model because the covariances between independent variables and covariances between mediators were significant, and constraining them to zero resulted in a significant mismatch between the data and the formulated model. Therefore, covariances between these variables had to be controlled. Fit statistics for these models are presented in the note for Table 2. In conclusion, the saturated model was the best fit because it perfectly reproduced all variances and covariances. The saturated model had a chi-square of zero with zero degrees of freedom and had the maximum values of the remaining fit statistics (Χ^2^(0) = 0.00; *CFI* = 1.00; *TLI* = 1.00; *NFI* = 1.00; *IFI* = 1.00; *RMSEA* = 0.00; 90%CI [00–0.00]; *PCLOSE* = 0.000; *SRMR* = 0.00; *GFI* = 1.00; *AGFI* = 1.00).

### 3.2. Estimates of Model Direct Path Value

Table 3 shows the direct path coefficient estimates. These results show that increased levels of FNAES, but not CAS, β = −0.10; *Z* = −1.86; *p* > 0.05, had significant influences on decreased well-being levels, β = −0.17; *Z* = −2.77; *p* < 0.01 (the subsequent analysis of indirect effects will show that these relations were mediated). Further analysis showed that increased levels of FNAES had significant influences on decreased levels of dieting, β = −0.48; *Z* = −10.62; *p* < 0.001, but CAS had an insignificant impact, β = −0.10; *Z* = −1.86; *p* > 0.05. Then, it was observed that increased levels of CAS and FNAES had significant influences on decreased levels of B&F preoccupation (β = −0.33; *Z* = −5.16; *p* < 0.001 and β = −0.39; *Z* = −7.83; *p* < 0.001, respectively) and decreased levels of oral control (β = −0.31; *Z* = −4.77; *p* < 0.001 and *β* = −0.27; *Z* = −5.27; *p* < 0.001, respectively). Thus far, we have observed that both independent variables (CAS and FNAES) had impacts on specified mediators (dieting, B&F preoccupation, and oral control), with one insignificant exception being the influence of CAS on dieting. The analysis of the final part of the model showed that increased levels of dieting had significant influences on decreased levels of well-being, β = −0.19; *Z* = −2.96; *p* < 0.01, but B&F preoccupation increased well-being, β = 0.33; *Z* = 4.35; *p* < 0.001. Oral control was not related significantly to well-being, β = 0.01; *Z* = 0.24; *p* > 0.05. These results mean that in the context of mediators, only dieting and B&F preoccupation influenced well-being. The results are also shown in Figure 1.

### 3.3. Estimate of Model Indirect Path Values

Table 4 shows the estimates of indirect effects. The results show that the relationship between FNAES and well-being was mediated by dieting, B&F preoccupation, and oral control. The same mediating patterns were observed in the relationship between CAS and well-being, but with one exception being the mediating effect of dieting. Dieting was not a mechanism in this relationship. Generally, the observed patterns of mediation effects mean that CAS and FNAES were negatively related to eating attitudes. More specifically, dieting was negatively related to well-being, but B&F preoccupation was related positively. Interestingly, oral control was also a mediating mechanism, but its direct effect on well-being was insignificant. Generally, with a few exceptions, it can be concluded that dieting attitudes are mechanisms that relate FNAES and CAS to well-being.

## 4. Discussion

In the current analysis, we searched for a model to explain the relations between certain psychological characteristics influencing the well-being of managers in Poland. We were particularly interested in the potentially significant roles of variables such as COVID-19 anxiety (measured with the coronavirus anxiety scale; CAS), fear of negative assessment of one’s physical appearance (measured with the fear of negative appearance evaluation scale; FNAES) in the workplace, as well as eating attitudes (dieting, bulimia and food preoccupation, and oral control behavior) as mechanisms underlying these relationships. The analysis revealed that increased fear of negative appearance, but not COVID-19 anxiety, significantly decreased the levels of well-being. However, the covariation between the FNAES and psychological well-being was not due to a direct cause-and-effect relationship. The model found that eating attitudes, such as dieting, bulimia, and food preoccupation, relate the fear of negative appearance evaluation and COVID-19 anxiety to well-being. Nevertheless, COVID-19 anxiety does not significantly influence the well-being levels of the studied respondents. It can be concluded that managers whose self-images are highly dependent on the assessments of other co-workers are characterized by lower well-being. However, only when they undertake healthy diets do they simultaneously express obsessive attention to food and bulimic behaviors. It is very important to take into account that eating attitudes in the current study were measured with the eating attitude test (EAT-26), which was originally developed on a clinical sample to estimate the tendency toward eating disorders; therefore, the results should be interpreted with caution [31].

When analyzing our model, it could be speculated that managers are an occupational group with leadership personalities who desire control and order. These personality aspects could be helpful when managing workplaces and managing healthy behaviors/eating restrictions. The landmark longitudinal studies of Mishel et al. [34] proved that self-control and the ability to delay gratification are strongly related to success in life. Some similar conclusions have motivated the new psychological concept by Grit [35]. Zbierowski and Gojny-Zbierowska [36], in their analysis of character strengths that contribute to entrepreneur success, emphasize the need for self-regulation, i.e., self-control. In difficult workplace situations, the most essential traits for success are persistence, self-regulation, humor, enthusiasm, teamwork, fairness, and leadership. Self-regulation could also help manage vices and habits, allowing managers to cope with pressure, impulses, and emotions. It seems that the studied managers could also compensate for their fear of negative appearance evaluations via food preoccupation and dieting. These results suggest that investments in appearance could motivate managers to diet in order to eliminate the fear of negative evaluations from others [37]. Furthermore, in a comparison between Poles and Indonesians, Novita et al. [23] found that Polish respondents more often expressed themselves through healthy dieting. Another possible explanation for the significant role of dieting and food preoccupation as a mediator between fear of negative appearance and well-being is that there is social pressure toward eating healthy and eating less. Nowadays, in many cultures, eating a healthy diet and paying attention to nutrition are considered important. Moreover, eating behaviors are within one’s control and can symbolize a manager’s internal locus of control. In a study on engineers, Rambe and Modise [38] found that the internal locus of control with the combination of behavior-focused strategies and self-leadership strategies had the most significant influence on job performance. On the other hand, dietary patterns may have changed during the lockdown and the pandemic, as people started working from home. Sorić et al. [39], in their study on Croatians during the COVID-19 pandemic, identified favorable changes toward better dieting that could be beneficial for physical and mental health.

Considering COVID-19 anxiety and well-being, we assumed that there would be a significant influence of emotions associated with COVID-19 on the levels of well-being. The results did not confirm our assumptions. This observation is similar to other studies that found that populations have adjusted to the pandemic for various reasons, including the availability of vaccination programs, reliable information on the consequences of the coronavirus disseminated through mass media, and established patterns of healthy behaviors [40].

This study has limitations that are typical of correlational studies. Investigating the model based on age and gender could generate valuable results. In addition, the limitations of the presented study include the recruitment procedure—the sampling was purposive, which does not meet the criteria of a random selection. Future studies could recruit managers from a greater variety of companies and residences. Moderators, such as job experiences and positions in the hierarchy of the company, could also be taken into account. Past disordered eating should also be investigated. Future studies should consider including the aforementioned variables and moderators as well as explore interventions that could benefit the physical and mental health of managers. Moreover, the mental health of employees could provide substantial information regarding effective crisis management strategies in the future.

## 5. Conclusions

Positive body images were associated with high levels of well-being among managers, but only when mediated by healthy dieting and paying attention to nutrition. While the well-being level of managers was high, it is worth investigating how they can better flourish and develop in life and work. The well-being of managers could transfer to the quality of life of their co-workers and companies. Nowadays, more jobs require creativity, which strongly correlates with positive emotions. In the long run, workers must be cheerful to be creative and innovative. Generating a positive work atmosphere is part of the responsibility of emotionally intelligent and happy leaders.

## Figures and Tables

**Figure 1 ijerph-20-00637-f001:**
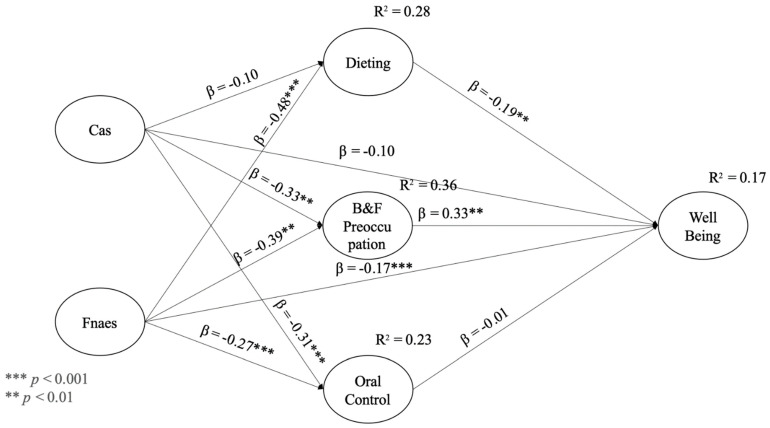
Estimate of model path values.

**Table 1 ijerph-20-00637-t001:** Zero-order correlations, reliability, and basic descriptive statistics.

Measure	Cronbach’s α	M	SD	1	2	3	4	5
Dieting (1)	0.85	4.53	0.77					
B&F preoccupation (2)	0.82	5.33	0.65	0.62 ***				
Oral control (3)	0.59	5.09	0.60	0.42 ***	0.39 ***			
Well-being (4)	0.75	4.60	0.47	0.14 *	0.28 ***	0.14 *		
CAS (5)	0.88	1.37	0.60	−0.26 ***	−0.35 ***	−0.30 ***	−0.25 ***	
FNAES (6)	0.93	2.10	0.89	−0.51 ***	−0.48 ***	−0.35 ***	−0.26 ***	0.33 ***

Legend: B&F—bulimia and food; CAS—COVID anxiety scale; FNAES—fear of negative appearance evaluation scale. *** *p* < 0.001; * *p* < 0.05.

**Table 2 ijerph-20-00637-t002:** Model fit comparisons.

Comparison	Models	DF	AIC	BIC	Χ^2^	Χ^2^ diff	DF diff
1	Saturated model	0	1613.87	1683.52	0.00		
	Model with constrained covariances between independent variables	1	5481.25	5558.63	54.57	64.34 ***	1
2	Saturated model	0	1613.87	1683.52	0.00		
	Model with constrained covariances between mediators	3	1759.33	1817.37	151.46	134.5 ***	3
3	Model with constrained covariances between independent var (*a*)	1	5481.25	5558.63	54.57		
	Model with constrained covariances between mediators (*b*)	3	1759.33	1817.37	151.46	76.59 ***	2

Note: Χ^2^ diff = differences between Χ^2^ estimates; *p* = *p*-value for Χ^2^; df = degrees of freedom; AIC = Akaike information criterion; BIC = Bayesian information criterion. *a* = Fit statistics: Χ^2^(1) = 27.56; *p* < 0.001; *CFI* = 0.95; *TLI* = 0.34; *NFI* = 0.95; *IFI* = 0.95; *RMSEA* = 0.27; 90%PU [0.19–0.37]; *PCLOSE* = 0.000; *SRMR* = 0.05; *GFI* = 0.97; *AGFI* = 0.32; *b* = Fit statistics: Χ^2^(3) = 151.46; *p* < 0.001.; *CFI* = 0.74; *TLI* = −0.22; *NFI* = 0.74; *IFI* = 0.74; *RMSEA* = 0.37; 90%PU [0.32–0.43]; *PCLOSE* = 0.000; *SRMR* = 0.10; *GFI* = 0.82; *AGFI* = −0.26. *** *p* < 0.001.

**Table 3 ijerph-20-00637-t003:** Estimates of model path values.

Dependent var	<-	Independent var	*B*	s.e.	*Z*	LCI	UCI	β
Well-being	<-	Cas	−0.08	0.04	−1.86	−0.16	0.00	−0.10
Well-being	<-	Fnaes	−0.09	0.03	−2.77 **	−0.15	−0.03	−0.17
Dieting	<-	Cas	−0.13	0.07	−1.86	−0.26	0.01	−0.10
Dieting	<-	Fnaes	−0.42	0.04	−10.62 ***	−0.49	−0.34	−0.48
B&F preoccupation	<-	Cas	−0.36	0.07	−5.16 ***	−0.5	−0.22	−0.33
B&F preoccupation	<-	Fnaes	−0.28	0.04	−7.83 ***	−0.35	−0.21	−0.39
Oral control	<-	Cas	−0.31	0.07	−4.77 ***	−0.44	−0.18	−0.31
Oral control	<-	Fnaes	−0.18	0.03	−5.27 ***	−0.25	−0.11	−0.27
Well-being	<-	Dieting	−0.11	0.04	−2.96 **	−0.19	−0.04	−0.19
Well-being	<-	B&F Preoccupation	0.24	0.06	4.35 ***	0.13	0.35	0.33
Well-being	<-	Oral Control	0.01	0.04	0.24	−0.08	0.10	0.01

Note: *B* = unstandardized regression coefficient; s.e. = standard error for *B*; *Z* = Z statistics; LCI and UCI = 95% confidence intervals (lower and upper, respectively); β = standardized regression coefficient. *** *p* < 0.001; ** *p* < 0.01.

**Table 4 ijerph-20-00637-t004:** Estimate of the model’s indirect effects.

Effect	*B*	s.e.	*Z*	LCI	UCI	β
CAS -> Dieting -> Well-being	0.01	0.01	1.53	0.00	0.03	0.02
FNAES -> Dieting -> Well-being	0.05	0.02	2.83 **	0.01	0.08	0.09
CAS -> B&F Preoccup -> Well-being	−0.09	0.03	−3.20 **	−0.14	−0.03	−0.11
FNAES -> B&F Preoccup -> Well-being	−0.07	0.02	−3.96 ***	−0.10	−0.03	−0.13
CAS -> Oral Control -> Well-being	−0.07	0.02	−3.07 **	−0.12	−0.03	−0.10
FNAES -> Oral Control -> Well-being	−0.04	0.01	−3.13 **	−0.07	−0.02	−0.09

Legend: B&F Preoccup—bulimia and food preoccupation; CAS—COVID anxiety scale; FNAES—fear of negative appearance evaluation scale. Oral Control—oral and control behavior. Note: *B* = unstandardized regression coefficient; s.e. = standard error for *B*; *Z* = Z statistic; LCI and UCI = 95% confidence intervals (lower and upper, respectively); β = standardized regression coefficient. *** *p* < 0.001 ** *p* < 0.01.

## Data Availability

Not applicable.

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
