# Peer review of "Well-Being of High-Level Managers during the Pandemic: The Role of Fear of Negative Appearance, Anxiety, and Eating Behaviors"

_ijerph, 2022, doi:10.3390/ijerph20010637_

Round 1

Reviewer 1 Report (Previous Reviewer 2)

I appreciate the changes that the authors have made to this draft, but feel that the initial problems still remain.  It is unclear why we should "care" about managers - how are they unique? why focus on them? - and key information about the study method is still incomplete.  There is no theory utilized, and the explanation that one does not exist for this context is not accurate nor is it an acceptable explanation.  Few additions have been made to this draft; much of the work has been basic copyediting.

Author Response

Cover letter to the reviewers.

We would like to thank every reviewer for making remarks concerning the improvement of our paper. We will address remarks in one-by-one answers.

Answer to the Reviewer 1.

I appreciate the changes that the authors have made to this draft, but feel that the initial problems still remain.  It is unclear why we should "care" about managers - how are they unique? why focus on them? - and key information about the study method is still incomplete.  There is no theory utilized, and the explanation that one does not exist for this context is not accurate nor is it an acceptable explanation.  Few additions have been made to this draft; much of the work has been basic copyediting.

We are sorry that our changes don’t satisfy your concerns. Based on the opinions of other reviewers (not only of this paper but the whole grant project) the justification of the theory and why we care about the managers was sufficient. Of course, we acknowledge that your concerns would improve the quality of this study, but we are not able to give further explanation.

Reviewer 2 Report (New Reviewer)

I was very pleased to read this article even if there is a big flaw in the way it is presented. The authors had submitted a work version of their work with the correction and remarks notes on. So it is very unpleasant to read. Nevertheless it will be correct very quickly just by on click.
That why I recommend this article for publication when this small error will be corrected.

Author Response

Cover letter to the reviewers.

We would like to thank every reviewer for making remarks concerning the improvement of our paper. We will address remarks in one-by-one answers.

Answer to the Reviewer 2.

I was very pleased to read this article even if there is a big flaw in the way it is presented. The authors had submitted a work version of their work with the correction and remarks notes on. So it is very unpleasant to read. Nevertheless it will be correct very quickly just by on click. That why I recommend this article for publication when this small error will be corrected.

We would like to thank you for such a positive review, we are sorry for this error, which of course has been corrected now.

Reviewer 3 Report (New Reviewer)

Your research is a good reflection of the trend.

I suggest a few things to make your research more clear to the reader.

1. Your article needs to mention the survey items. In this study, it is important to check the content of the questions because the answers may appear differently depending on the content of each question. Statistical results cannot be trusted if objectification is not completed for the survey items. In addition, since this paper grouped the items and checked the relationship between the groups, it is necessary to verify whether each item can represent the phenomenon.

Ex) In the case of B&F preoccupation, it should be possible to define whether the B&F preoccupation is a serious situation or can be said to have no problem according to the answer level of the question.

2. In the paper, the impact of COVID-19 on leaders was confirmed, but no special results were confirmed about the impact on the employees they manage. If your research is a survey of group managers, you need results about the relationship or method with employees perceived by the surveyor.

3. Why were the EAT-26 items divided into 3 items for analysis? Other items are also grouped, but not divided into three. And since your article is about checking Well-being, CAS, and FNAES based on EAT-26, there should be an explanation of EAT-26 in the description.

4. In order to abbreviate a sentence, the modification of the notation to an abbreviation must be explained somewhere.

Are the 3 subscales of EAT-26 'dieting', 'bulimia', and 'food preoccupation'?

Or is it ‘dieting’, ‘bulimia and food preoccupation’, or ‘oral and control behavior’?

If it is the latter, please rewrite the 5th paragraph of 2.2 instrument as a whole.

The way you've written it doesn't confirm that B&F preoccupation is bulimia and food preoccupation, and oral control is oral and control behavior.

5. There is a part to be modified minorly.

Figure 1. Fnaess -> Fnaes

Author Response

Cover letter to the reviewers.

We would like to thank every reviewer for making remarks concerning the improvement of our paper. We will address remarks in one-by-one answers.

I suggest a few things to make your research clear to the reader.

  1. Your article needs to mention the survey items. In this study, it is important to check the content of the questions because the answers may appear differently depending on the content of each question. Statistical results cannot be trusted if objectification is not completed for the survey items. In addition, since this paper grouped the items and checked the relationship between the groups, it is necessary to verify whether each item can represent the phenomenon.

Ex) In the case of B&F preoccupation, it should be possible to define whether the B&F preoccupation is a serious situation or can be said to have no problem according to the answer level of the question.

We underlined in the Discussion paragraph that: „It is very important to take into account that eating attitudes in the current study were measured with the Eating Attitude Test (EAT-26), which was originally developed on a clinical sample to estimate the tendency towards eating disorders; therefore, the results should be interpreted with caution”.

Moreover, while clinically interpreting the results of the EAT-26, the total score of 20 points and above means that the respondent should seek the advice of a qualified mental health professional. However, we analyzed 3 separate subscales as mediators. Furthermore, the authors of the EAT-26 do not give the interpretation of the results for each scale separately.

Furthermore, we were interested in elaborate relationships between variables and their mechanisms (mediators). So diagnostics aspects or normalization issues in our study are not the main concerns. We believe, that this is crucial aspect in clinical or screening studies.

  1. In the paper, the impact of COVID-19 on leaders was confirmed, but no special results were confirmed about the impact on the employees they manage. If your research is a survey of group managers, you need results about the relationship or method with employees perceived by the surveyor.

We follow the reviewer’s comment that the results on the impact of Covid-19 pandemic on employees would be worth studying. However, we focused on the managers’ characteristics in this particular study and we do not have their employees' results. Therefore we added this comment as a limitation of the study. (line….

  1. Why were the EAT-26 items divided into 3 items for analysis? Other items are also grouped, but not divided into three. And since your article is about checking Well-being, CAS, and FNAES based on EAT-26, there should be an explanation of EAT-26 in the description.

While the Wellbeing scale, as well as CAS and FNAES, interested us as overall scores, the EAT-26 had to be divided into 3 subscales to accurately measure which subscale is the significant mediator between CAS, FNAES, and Wellbeing. Thus, we were interested in how the particular specific explanatory eating mechanisms mediate between the predicted more general causes and outcomes.

  1. In order to abbreviate a sentence, the modification of the notation to an abbreviation must be explained somewhere.

Are the 3 subscales of EAT-26 'dieting', 'bulimia', and 'food preoccupation'?

Or is it ‘dieting’, ‘bulimia and food preoccupation’, or ‘oral and control behavior’?

If it is the latter, please rewrite the 5th paragraph of 2.2 instrument as a whole.

The way you've written it doesn't confirm that B&F preoccupation is bulimia and food preoccupation, and oral control is oral and control behavior.

Thank you for that comment. We clarified the description of the EAT-26 intrument: „The test consists of three subscales concerning: 1) dieting, 2) bulimia and food      preoccupation (B&F preoccupation) , and 3) oral and control behavior (Oral Control) (…)”

  1. There is a part to be modified minorly.

Figure 1. Fnaess -> Fnaes

The amendment has been applied.

Round 2

Reviewer 3 Report (New Reviewer)

Dear Author

I am grateful to the author for revising the study to reflect my review.

I have confirmed that you have sincerely responded to the issues I raised and reflected the contents.

Thank you for giving me the opportunity to review your good research.

This manuscript is a resubmission of an earlier submission. The following is a list of the peer review reports and author responses from that submission.

Round 1

Reviewer 1 Report

I found the article interesting and well-written. I would only suggest (before publication) informing more about the limitations of the study and which could be the future steps for researchers. That is, I agree with the authors that a cross-sectional study may limit the overly scientific soundness of the study, however, there may be additional limitations regarding the sample, how you recruit participants etc. Please, clarify this aspect and propose future directions to emulate your study and propose novel prospects.

One more thing that I suggest is to strengthen the clarity of the paper via its structure. That is, I suggest the authors realize an overview paragraph at the end of the introduction where they

a) state the aim of the paper,

b) the organization of the paper (e.g., the first section reports the method, the second section the results etc..) and

c) explaining why this methodology is the most recommended.

With this paragraph, the readers will have a more clear idea of the contents of the paper and what they will read in the following sections.

Author Response

Response to reviewer 1

Thank you for reviewing our paper and giving insights about how to improve our work. We hope that our answers will be adequate to your suggestions.

Thank you for advice on adding a paragraph with overview, we did it at the end of introduction chapter, with clarifying aim of the study and organization of the paper (Lines 81-102).

We have also added Research question (according to suggestion of the second reviewer) which in our opinion makes it clearer why this kind of methodology was used in our study (Lines 98-100).

Reviewer 2 Report

Thank you for the opportunity to read, "Well-being of High-level Managers During Pandemic."  The topic of this paper was interesting to me, as someone who studies occupational sociology and workplace environments.

Although the paper was of interest, there are a number of ways in which the manuscript itself can be significantly strengthened:

1. The introduction related to the pandemic itself is useful, but the rest of the material about Covid-related stressors is not unique to managers.  Concerns about health, working from home, sedentary behaviors... these were issues faced by many professional employees who were able to remain employed.  The few things that the authors did point out that were unique to managers - responsibility for changes at work, supervising others - were noted in a single sentence.  Why this sample?  Why focus on managers?

2. What is the theory driving this study?  This should be evident and detailed, as it holds together the paper and determines the variables chosen and hypotheses.

3. Greater detail about the sampling frame, how companies and managers were identified, which industries, etc., is needed in order to both determine merit of the method and better contextualize the results.  For instance, how was "manager" defined?  Was this study related to one type of industry (e.g., retail or healthcare?)?  What was the response rate?

4. As pertains to the scales, please identify results from factor analyses and provide sample items.

5. It is hard to adequately review the results without a theory and hypotheses.  Given that no hypotheses were presented - even implicitly, a reader cannot determine whether the statistical choices were appropriate.

6. The discussion section suffers from a great deal of speculation and brings in a number of concepts that have not been measured.  More care should be taken with the language as well - statistical models cannot "prove" findings.  There are a number of alternate explanations that are far simpler than those provided by the authors.  For instance, could it also be that work from home changes food access and dietary patterns?  I was surprised, given the topic, that the authors did not control for past disordered eating.

7. Throughout, copyediting changes are needed.  Many sentences can be simplified and more focused for ease of reading and better understanding.  There are unnecessary articles throughout the paper ("the" workplaces) and in other places, missing articles.  

I hope the authors find this feedback useful in developing their work.

Author Response

Response to Reviewer 2

Thank you for reviewing our paper and giving insights about how to improve our work. We hope that our answers will be adequate to your suggestions.

  1. The introduction related to the pandemic itself is useful, but the rest of the material about Covid-related stressors is not unique to managers. Concerns about health, working from home, sedentary behaviors... these were issues faced by many professional employees who were able to remain employed. The few things that the authors did point out that were unique to managers - responsibility for changes at work, supervising others - were noted in a single sentence.  Why this sample?  Why focus on managers?

Thank you for this suggestion. We have added additional chapter (according to the suggestion of Second Reviewer) in which we emphasize the role of managers (Lines 85-95)

We have added additional paragraph with managers as role models in creating healthy lifestyle (59-63).

  1. What is the theory driving this study? This should be evident and detailed, as it holds together the paper and determines the variables chosen and hypotheses.

Based on assumption that pandemic was unprecedented and that previous scientific research has not resulted in a coherent theory of how high-level workers respond to this situation, this study is exploratory. We have determined variables on base of another exploratory research made by:

 Wang, R.; Gan, Y.; Wang, X.; Li, J.; Lipowska, M.; Izydorczyk, B.; Guo, S.; Lipowski, M.; Yang, Y.; Fan, H. The Mediating Effect of Negative Appearance Evaluation on the Relationship Between Eating Attitudes and Sociocultural Attitudes Toward Appearance. Frontiers in psychiatry 2022, 13, 776842, doi:10.3389/fpsyt.2022.776842.

And Novita, S.; Andriani, D.; Erika; Lipowski, M.; Lipowska, M. Anxiety towards COVID-19, Fear of Negative Appearance, Healthy Lifestyle, and Their Relationship with Well-Being during the Pandemic: A Cross-Cultural Study between Indonesia and Poland. Int J Environ Res Public Health 2022, 19, doi:10.3390/ijerph19127525.

  1. Greater detail about the sampling frame, how companies and managers were identified, which industries, etc., is needed in order to both determine merit of the method and better contextualize the results. For instance, how was "manager" defined? Was this study related to one type of industry (e.g., retail or healthcare?)?  What was the response rate?

The managers were mainly from IT and insurance companies. It was hard to determine response rate due to the fact that the questionnaire was performed online. Manager was defined as a person managing at least 10 employees.

  1. As pertains to the scales, please identify results from factor analyses and provide sample items.

The used scales are validated in multiple studies, but we can conduct confirmatory factor analyses to confirm factor validity. Sample items were provided (Lines 142-150)

  1. It is hard to adequately review the results without a theory and hypotheses. Given that no hypotheses were presented - even implicitly, a reader cannot determine whether the statistical choices were appropriate.

Research question was added to determine statistical choice (Lines 96-100).

  1. The discussion section suffers from a great deal of speculation and brings in a number of concepts that have not been measured. More care should be taken with the language as well - statistical models cannot "prove" findings. There are a number of alternate explanations that are far simpler than those provided by the authors.  For instance, could it also be that work from home changes food access and dietary patterns?  I was surprised, given the topic, that the authors did not control for past disordered eating.

We changed the sound of Discussion in selected text parts, lines: 248. We supported the Discussion with the recommended explanation about dietary patterns, lines 273-277. We wrote about the lack of control for past disordered eating in the study's limitations, line 289.

  1. Throughout, copyediting changes are needed. Many sentences can be simplified and more focused for ease of reading and better understanding. There are unnecessary articles throughout the paper ("the" workplaces) and in other places, missing articles. 

We checked the text for missing articles.

I hope the reviewers finds our answers sufficient and satisfying.